# Antibiotic Prescribing for Lower Respiratory Tract Infections and Community-Acquired Pneumonia: An Italian Pediatric Emergency Department’s Real-Life Experience

**DOI:** 10.3390/life13091922

**Published:** 2023-09-15

**Authors:** Luca Pierantoni, Valentina Lasala, Arianna Dondi, Marina Cifaldi, Ilaria Corsini, Marcello Lanari, Daniele Zama

**Affiliations:** 1Pediatric Emergency Unit, IRCCS Azienda Ospedaliero-Universitaria di Bologna, 40138 Bologna, Italy; luca.pierantoni@aosp.bo.it (L.P.); ilaria.corsini2@unibo.it (I.C.); marcello.lanari@unibo.it (M.L.); daniele.zama2@unibo.it (D.Z.); 2Specialty School of Paediatrics, Alma Mater Studiorum, University of Bologna, 40126 Bologna, Italy; valentina.lasala@studio.unibo.it; 3Department of Medical and Surgical Sciences, Alma Mater Studiorum, University of Bologna, 40126 Bologna, Italy; 4Pediatric Clinic, Department of Medicine and Surgery, Azienda Ospedaliero-Universitaria of Parma, 43126 Parma, Italy; marina.cifaldi@gmail.com

**Keywords:** lower respiratory tract infection, community-acquired pneumonia, bronchiolitis, bronchitis, antibiotic prescription, pediatric emergency department, guidelines

## Abstract

Background: Lower respiratory tract infections (LRTIs) and community-acquired pneumonia (CAP) are among the most frequent reasons for referrals to the pediatric emergency department (PED). The aim of this study is to describe the management of antibiotic prescription in febrile children with LRTI or CAP admitted to a third-level PED and to investigate the different variables that can guide physicians in this decision-making. Methods: This is an observational, retrospective, monocentric study including patients < 14 years old who were presented to the PED for a febrile LRTI or CAP during the first six months of the year 2017. Demographic and clinical data, PED examinations, recommended therapy, and discharge modality were considered. Two multivariate logistic regression analyses were performed on patients with complete profiles to investigate the impact of demographic, laboratory, and clinical variables on antibiotic prescription and hospital admission. Results: This study included 584 patients with LRTI (*n* = 368) or CAP (*n* = 216). One hundred and sixty-eight individuals (28.7%) were admitted to the hospital. Lower age, higher heart rate, and lower SpO2 were associated with an increased risk of hospitalization. Antibiotics were prescribed to 495 (84.8%) patients. According to the multivariate logistic regression, the diagnosis and duration of fever were substantially linked with antibiotic prescription. Conclusions: The present study reports real-life data about our PED experience. A high rate of antibiotic prescription was noted. In the future, it is necessary to improve antibiotic stewardship programs to increase clinical adherence to guidelines.

## 1. Introduction

Lower respiratory tract infections (LRTIs) are among the most common reasons for pediatric emergency department (PED) visits [1]. Among LRTIs, community-acquired pneumonia (CAP) is a relevant cause of pediatric death worldwide, accounting for more than 900,000 deaths among children younger than 5 years of age in 2015. Although the rate of mortality due to CAP is much lower in the developed world compared with the developing world, CAP continues to account for a significant proportion of health care visits and hospitalizations in high-income countries [2]. In infants and young children, LRTIs are mostly caused by viruses and are generally mild and self-limiting [1]. Despite their debatable beneficial effects, antibiotics are routinely over-prescribed in children with viral LRTI, with prescription rates ranging from 24 to 87% in Europe [1,3]. This high and variable rate is likely due to concerns about children’s susceptibility to bacterial infections, possible development of secondary complications, parental request, or physician uncertainty about the causative pathogen [4,5,6]. However, antibiotic overuse is a well-known cause of unnecessary side effects that may promote the development of antimicrobial resistance [7].

Antibiotic resistance is an increasingly serious public health problem; a clear relationship was observed between primary care antibiotic prescribing (80% of all antibiotic prescribing) and antibiotic resistance [8,9].

With regard to resistance patterns in the pediatric population in Italy, the 2020 Emilia-Romagna regional report showed a rate of resistance to amoxicillin for Streptococcus pneumoniae equal to 8.3% [10]. Moreover, in the 2021 Emilia-Romagna regional report the antibiotic resistance data referring to Escherichia coli isolates from urine cultures showed a reduction in antibiotic resistance trends probably due to the lower antibiotic prescription during the COVID-19 pandemic. In fact, antibiotic prescription in children in the Emilia-Romagna region showed a significant reduction in all classes of antibiotics and all age groups starting from 2013, with a partial recovery in 2018–2019, followed by a clear decline in 2020–2021 during the COVID-19 pandemic. In 2021, an increase in prescriptions was observed (+64%) compared to 2020 (822 prescriptions/1000 child-years in 2019; 404.9 in 2020; 390.8 in 2021; 639.4 in 2022). Amoxicillin is the most prescribed antibiotic in Emilia-Romagna for pediatric patients, having surpassed penicillin associated with beta-lactamase inhibitors since 2016 [11].

All physicians should always follow guidelines when selecting whether and which antibiotics to administer [12,13]. For example, the British National Institute for Health and Care Excellence (NICE) guidelines concur that antibiotics should not be routinely administered in patients with bronchiolitis, the most common LRTI during the first year of life [14]. With regard to CAP, the Infectious Diseases Society of America (IDSA) guidelines do not recommend routine antibiotic therapy in preschool children due to the predominantly viral etiology [15]. Viruses are the main responsible agents for CAP in children under the age of two, with Respiratory Syncytial Virus (RSV) being the most common.

Bacteria, particularly *S. pneumoniae*, are identified as responsible for CAP as children get older [15,16,17]. A combined viral–bacterial infection occurs in 20–30% of pediatric CAP cases [16,17]. Amoxicillin should be the first-line treatment in preschool-age children with mild–moderate CAP of suspected bacterial origin as well as in school-age children and adolescents with *S. pneumoniae* CAP. Macrolides should be given to school-age children and adolescents with CAP due to atypical bacteria [15]. It is still difficult to distinguish between pediatric patients who will benefit from antibiotic treatment and those who will not. A greater understanding of the factors influencing and driving antibiotic prescription decisions, in particular, is critical for future treatment strategies [18].

The aim of this study is to describe the management of antibiotic prescription in children with LRTI admitted to a third-level PED and to investigate the different variables that can help physicians make these decisions.

## 2. Materials and Methods

### 2.1. Study Design

This is an observational, retrospective, monocentric study including patients younger than 14 years of age who were referred to the PED of the IRCCS—AOU Policlinico di S.Orsola, Bologna (Italy), between 1 January 2017 and 30 June 2017. Inclusion criteria were: (1) body temperature > 37.5 °C (measured at the PED or reported by parents); and (2) diagnosis of LRTI or CAP defined at discharge. We considered the aforementioned fever threshold to keep the sample more homogeneous.

Our center is an urban, academic, tertiary care facility, consisting of a PED with approximately 24,000 visits per year of children aged 0–14, a short-stay observation unit with 6 beds, and a ward with 28 beds. Patients referred to the PED are registered by a triage-qualified nurse; two more nurses are available for the PED and one for the short-stay observation patients. The medical visit is performed by both one or more pediatric residents and the attending physician. For each patient, a record form is filled in by the nurse and medical staff, including demographic data, anamnestic information, vital signs, physical examination, any performed treatments, subsequent clinical reevaluations, diagnosis, discharge modality (home or admission to the ward), and home therapy prescription when applicable. For children requiring a short-stay observation, prescribed examinations, therapy, and subsequent reevaluations are also noted on the same form.

The diagnoses of LRTI or CAP defined at discharge were made considering clinical criteria used in daily practice. LRTIs included bronchiolitis, defined as an acute infection of the lower airways affecting infants under two years of age and characterized by respiratory distress, persistent cough, and wheezing or crackles on auscultation of the chest (or both); and bronchitis, defined as a self-limiting inflammation of the trachea and bronchi characterized by a persistent cough. CAP was defined as the presence of signs and symptoms of pneumonia including fever > 38.5 °C associated with chest recessions, new onset of cough, chest pain, and a raised respiratory rate in a previously healthy child due to an infection acquired outside the hospital.

The study staff reviewed the medical records to collect demographic and clinical data (age, gender, weight, heart rate, respiratory rate, peripheral oxygen saturation, number of days of fever duration, therapy administered before the visit to PED), investigations performed in the PED (laboratory tests and/or chest X-ray), outcomes (home discharge or hospital admission), and prescribed antibiotic therapy (both at home discharge and during hospital stay), which included the type of molecule, the relative dose in mg/kg/day, the number of daily administrations, and the days of treatment duration.

Patients with incomplete medical records that did not allow for the confirmation of a diagnosis of LRTI or CAP, as well as those who were discharged from the PED against medical advice before completing the evaluation were excluded (Figure 1).

### 2.2. Statistical Analysis

Parametric continuous variables were reported as means and standard deviation (SD). Nonparametric continuous variables were expressed as the median and interquartile range (IQR). Categorical variables were expressed as numbers and relative percentages. Patients were compared according to their diagnostic class (LRTI or CAP) and whether they were admitted to the hospital or discharged from the PED. Pearson’s 𝛘2 test was used to compare the qualitative variables. Kolmogorov–Smirnov was used to assess the quantitative variables for normal distribution. Parametric and nonparametric variables were compared using the *T*-Test or Mann–Whitney test, respectively. Finally, we performed logistic regression analysis to assess the impact of demographic, laboratory, and clinical factors on antibiotic prescription. To eliminate confounding effects, all independent variables with a statistically significant impact on the dependent variable were suddenly included in a multivariate logistic regression. Similarly, univariate and multivariate logistic regressions with hospital admission as the dependent variable were carried out. *p* values < 0.05 were considered statistically significant. The Statistical Package for the Social Sciences (SPSS-version 17.0) software was used for the analysis of data. The data was collected anonymously as part of a clinical audit carried out in the interest of improving the quality of health care, in agreement with the representatives of the referenced health facility; therefore, it was not submitted to the opinion of the Ethics Committee.

## 3. Results

We enrolled 584 patients with a diagnosis of LRTI (*n* = 368, 63%) or CAP (*n* = 216, 37%) out of 3160 patients assessed in our PED for a febrile illness throughout the study period. A hundred and sixty-eight (28.7%) patients were hospitalized. Table 1 summarizes the main clinical and demographic characteristics of the study cohort.

In this study, 495 (84.8%) of 584 participants diagnosed with LRTI or CAP received an antibiotic prescription. We evaluated antibiotic prescriptions in the different age groups (under 2 years, 2–5 years, and over 5 years), in patients with LRTI or CAP, and in patients discharged home or hospitalized. We also investigated the type of antibiotic prescribed in the general population and in different patient groups (LRTI or CAP, discharged home or hospitalized). Table 2 reports this information.

Multivariate logistic regression analysis, performed considering hospital admission, showed a statistically significant correlation between hospital admission and the following variables: heart rate > 160 bpm (OR 2.388, CI 95% 1.365–4.179, *p*-value 0.002), peripheral oxygen saturation < 92% (OR 3.119, CI 95% 1.050–9.266, *p*-value 0.041), age 2–5 years (OR 0.369, CI 95% 0.199–0.684, *p*-value 0.002), age > 5 years (OR 0.394, CI 95% 0.164–0.948, *p*-value 0.038), as reported in Table 3.

Finally, diagnosis of LRTI (OR 0.069, CI 95% 0.016–0.267, *p*-value < 0.001) and >3 days of fever (OR 3.633, CI 95% 1.752–7.533, *p*-value < 0.001) were significantly correlated to any antibiotic prescription in the multivariate logistic regression analysis (Table 4).

## 4. Discussion

This study examines the real-life management of antibiotic prescription in children admitted to a third-level PED with LRTI and CAP. Among the main findings, our center revealed one of the highest antibiotic prescription rates in Europe and a tendency to employ second-line medications, despite the widespread adoption of worldwide pediatric recommendations for LRTI and CAP [14,15,17].

Lower age, higher heart rates, and lower peripheral oxygen saturation were found to be related to a higher likelihood of hospitalization. This is consistent with the NICE guidelines for bronchiolitis, which indicate low peripheral oxygen as a major predictor of hospitalization [14]. Regarding pneumonia, IDSA guidelines identify younger age (lower than 3–6 months) and hypoxemia as risk factors for pneumonia severity and hospitalization, in agreement with our findings [17].

In our center, the global rate of antibiotic prescription was 84.8%. In particular, in patients with CAP, it reaches 97.2%, potentially in line with the guidelines recommendation, especially in children > 2 years old [17], while in patients with acute bronchitis and bronchiolitis, antibiotic prescription is 77.4%, despite the fact that guidelines do not recommend routine use of antibiotics in these conditions [14].

In Europe, the antibiotic prescription rate for LRTI is highly variable between countries, ranging from 24% to 87% in the pediatric population referred to the PED for fever and without comorbidities. Thus, our data confirm the high trend of antibiotic prescription in Italian hospitals, which is among the highest in Europe [3]. A study in the Netherlands about antibiotic prescription guidelines adherence in the pediatric population with fever observed that the prescription rate of general practices was 60% in CAP and from 23.2% to 70.1% in acute bronchitis [13].

The high rate of antibiotic prescription in the pediatric population is based on the well-known concerns regarding children’s susceptibility to bacterial infections and to the risk of secondary complications [4]. Furthermore, in an Italian study conducted on a population of both pediatricians and parents, it was observed that the important determinants of antibiotic prescription are etiology uncertainty and parental expectation of receiving these drugs [19].

About the type of antibiotic prescribed, according to the IDSA guidelines, the first-line antibiotic for the treatment of mild and moderate CAP is amoxicillin, while macrolides are used for the treatment of pneumonia from atypical pathogens in school-aged children and adolescents. In hospitalized patients, intravenous ampicillin is recommended as the first line, while third-generation parenteral cephalosporins should be used in regions where the local epidemiology of invasive pneumococcal strains documents high-level penicillin resistance [15]. In Italy, according to the Italian National Institute of Health’s report, the percentage of isolates of *S. pneumoniae* resistant to penicillin is 9.7%; thus, our country is not among the regions at high risk of resistance [20].

Amoxicillin was prescribed in about 60% of pediatric CAP cases in a Dutch research study, while other antibiotics, not recommended by the guidelines, were prescribed in 20% [13]. Similarly, in a trial of antibiotic therapy in pediatric CAP patients in the United States, 40.1%, 42.5%, and 16.8% were given amoxicillin, macrolides, and other broad-spectrum antibiotics (cephalosporins, amoxicillin combined with clavulanic acid), respectively [21].

Similarly to other studies, we found a trend towards prescribing second-line rather than first-line antibiotics: amoxicillin was provided to just 92 (25.2%) of the patients discharged home [13,21].

Nowadays, one of the main challenges in pediatrics is to recognize which patients will benefit from antibiotic treatment and those who will not; thus, it is critical to understand what factors influence antibiotic prescription. We attempted to investigate demographic, clinical, and laboratory drivers of antibiotic prescription in LRTIs and CAP. We observed that diagnosis and a longer duration of fever (>3 days) were statistically correlated with antibiotic prescription. Diagnosing LRTI reduces antibiotic prescription, which is consistent with the British National Institute for Health and Care Excellence (NICE) guidelines stating that antibiotics should not be used routinely in bronchiolitis [14].

In agreement with our study, there is an important double-blind, randomized, placebo-controlled trial published in 2021 which showed that antibiotics (amoxicillin) are unlikely to make a clinically important difference to the symptom burden for uncomplicated LRTI (non-pneumonic) in children—both overall and for the key clinical subgroups (patients with chest signs, fever, physician rating of unwell, sputum or chest rattle, and shortness of breath) where antibiotic prescribing is most common [8].

In terms of CAP, both the USA and British guidelines state that age should be one of the key determining variables for antibiotic prescription owing to the predominance of viral etiology in children under the age of two [15,17].

Several studies have been conducted to identify the factors that influence antibiotic prescription in pediatric patients with respiratory infections. Among these, a study performed on febrile children referred to emergency departments across Europe reported that, similar to our findings, a longer fever duration was correlated with antibiotic prescription. In contrast to our findings, an association with antibiotic prescription was also detected for focal or diffuse abnormalities in chest X-rays, higher CRP values, and an older age [3].

Another study on this topic evaluated a cohort of pediatric patients with LRTI who were referred to the PED and found that CRP, white blood cell count, body temperature, and pleural pain were all associated with antibiotic prescription [18]. Other studies have found an association between increased CRP levels and bacterial etiology in children with LRTI [22,23], despite the fact that CAP guidelines in the United States and in the United Kingdom do not recommend this laboratory test as a discriminator for bacterial or viral etiology [15,17]. Higher CRP levels were not related to antibiotic prescription in our trial, which followed the same parameters as the US and UK CAP.

According to the World Health Organization, pneumonia is diagnosed in all children 2–59 months old who have tachypnea and/or chest retractions. They should be treated at home for five days with oral amoxicillin and, in areas of low HIV prevalence and with limited resources, the duration of treatment for ‘fast breathing pneumonia’ can be decreased to three days [24]. However, the majority of children with ‘fast breathing pneumonia’ have viral, mild, and self-limiting illness [25]. Isolated tachypnea may not be sensitive or specific enough to diagnose bacterial pneumonia, making antibiotic treatment unnecessary [26,27]. In fact, in our analysis, respiratory rate had no statistically significant relationship with antibiotic prescription. Although one clinical feature is not sufficient to definitively diagnose bacterial pneumonia, a combination of clinical features may improve diagnostic performance [26]. Some studies show that combining clinical symptoms and blood biomarkers can improve discrimination between bacterial or viral CAP in children. For example, high CRP with fever increased specificity in distinguishing bacterial and viral pneumonia when compared to elevated CRP alone [28,29].

The inclusion of a large number of pediatric patients and the collection of many variables and data, reporting a real-life experience regarding the characteristics of the antibiotic prescription in an Italian third-level PED, is a major strength of our study. Furthermore, identifying two determinants of antibiotic prescription such as diagnosis and duration of fever of more than 3 days can help clinicians decide whether to prescribe antibiotic therapy.

The main limitation is that this is a single-center study; it would be interesting to examine the antibiotic prescription rates of various Italian hospital centers. Furthermore, patient data were obtained retrospectively and not all patients were included in the multivariate logistic regression analysis because of missing data.

A further limitation of this study is that it examined a time interval for enrollment without first determining the sample size required for the investigation. Because of the disparity between LRTI and non-LRTI, the high number of patients may have inflated the results. However, we think that a real-life study also has the advantage of documenting what happens daily in our reality so that we can plan improvement strategies.

In addition, we do not consider other variables investigated in the literature such as the presence of comorbidities, which has been associated with higher antibiotic prescription [30], the use of rapid diagnostic tests for viruses, which aid in reducing the performance of blood tests and chest X-rays and thus the use of antibiotics [12], and parental expectancy, which has also been associated with higher antibiotic prescription [19].

About the prescription of second-line antibiotics, it is unclear how many of these agents were appropriate because we did not record the individual resistances and allergies of patients.

## 5. Conclusions

In conclusion, our study analyzes the antibiotic prescription rate in an Italian third-level pediatric hospital, focusing on the type of molecules prescribed and the determinants of prescription. We confirmed a high prevalence of antibiotic prescription and a preference for second-line options, most likely as a result of poor adherence to guidelines, parental expectation, and difficulty discriminating between viral or bacterial LRTI. Staff training, antibiotic stewardship, periodic review of guidelines, and implementation of the use of rapid diagnostic viral tests are potential future interventions that may be introduced to lower the high rate of antibiotic prescription in our center. In the future, it will be a requirement to strengthen strategies such as training programs to boost clinical adherence to guidelines as well as to uncover more specific drivers of antibiotic prescribing. Our study adds to the fact that real-life studies frequently reveal non-adherence to guidelines, implying that stewardship programs for measures to promote clinical adherence to guidelines are required.

## Figures and Tables

**Figure 1 life-13-01922-f001:**
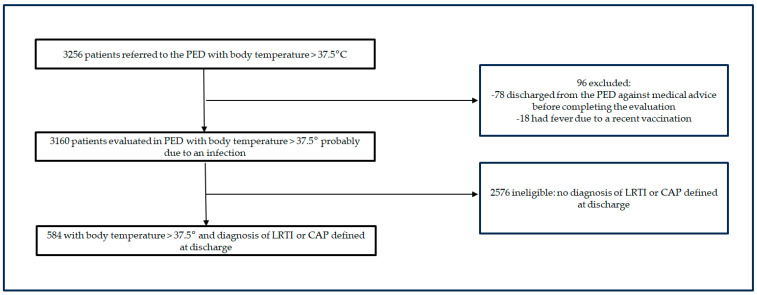
Study design.

**Table 1 life-13-01922-t001:** Characteristics of the study cohort compared by low respiratory tract infections (LRTI) and community-acquired pneumonia (CAP) and by hospitalized and home-discharged patients. CRP = C-Reactive Protein, SD = standard deviation, SpO_2_ = saturation of peripheral oxygen, WBC = white blood cells, CXR = chest X-ray, n.s. = not significant.

Variables	Total Population	LRTI *n* = 368	CAP *n* = 216	*p*	Patients Discharged Home *n* = 416	Patients Hospitalized *n* = 168	*p*
Females—*n* (%)	274 (46.9)	174 (47.5)	100 (46.5)	n.s.	220 (47.1)	54 (46.2)	n.s.
Blood test performed—*n* (%)	147 (25.2)	86 (23.5)	61 (28.4)	n.s.	96 (20.6)	51 (43.6)	<0.001
CRP > 3 mg/dL—*n* (%)	52 (8.9)	26 (7)	26 (12)	n.s.	18 (4.3)	34 (20.2)	n.s.
CRP (DS) [mg/dL]	4.7 (6.6)	3.2 (3.9)	6.8 (8.8)	0.004	3.3 (4.3)	5.2 (7.3)	0.026
CXR performed—*n* (%)	329 (56.6)	154 (42.1)	175 (81.4)	<0.001	267 (57.2)	64 (54.7)	n.s.
Consolidations at CXR—*n* (%)	113 (34.3)	8 (5.2)	105 (60.0)	<0.001	86 (32.2)	28 (43.8)	n.s
Mean age (SD) [months]	39.4 (35.7)	33.8 (32.9)	49.0 (38.3)	<0.001	42.5 (35.0)	27.2 (35.9)	<0.001
Mean weight (SD) [Kg]	15.6 (9.9)	14.1 (8.6)	17.7 (11.4)	<0.001	16.5 (9.9)	12.1 (9.7)	<0.001
Mean duration of fever (SD) [days]	3.1 (2.6)	2.9 (2.6)	3.4 (2.5)	0.012	3.0 (2.3)	3.3 (3.5)	n.s.
Mean body temperature (SD) [°C]	37.8 (1)	37.8 (1)	38.9 (1.1)	0.039	37.8 (1.1)	37.9 (1.1)	n.s.
Mean heart rate (SD) [bpm]	143.2 (22.9)	143.8 (23.0)	142.3 (23.0)	n.s.	140.4 (22.2)	154.5 (22.8)	<0.001
Mean respiratory rate (SD) [breaths per min]	39.9 (14.5)	41.9 (14.6)	35.8 (13.4)	<0.001	38.2 (13.8)	45.2 (15.1)	<0.001
Mean SpO2 (SD) [%]	96.6 (4.4)	96.8 (2.5)	96.3 (6.4)	n.s.	96.9 (2.1)	95.1 (8.9)	0.025
Mean WBC count (SD) [/mmc]	15.0 (7.8)	14.0 (6.7)	16.3 (8.9)	n.s.	13.9 (7.3)	13.9 (6.9)	n.s.
Mean neutrophil count (SD) [%]	63.4 (18.1)	59.22 (19.4)	68.7 (14.7)	<0.001	62.9 (17.3)	62.6 (71.6)	n.s.

**Table 2 life-13-01922-t002:** Antibiotic prescription in the two subpopulations of patients diagnosed with low respiratory tract infections (LRTI) and community-acquired pneumonia (CAP) and in the two subpopulations of hospitalized and home-discharged patients. PED = pediatric emergency department, IQR = interquartile range, n.s. = not significant.

	Total Population *n* = 584	LRTI *n* = 368	CAP *n* = 216	*p*	Patients Discharged Home *n* = 416	Patients Hospitalized *n* = 168	*p*
Antibiotic prescription—*n* (%)							
Overall	495 (84.8)	285 (77.4)	210 (97.2)	<0.001	365 (87.7)	130 (77.4)	<0.001
<2 years	190 (38.4)	130 (45.6)	60 (28.6)		118 (32.4)	72 (55.4)	
2–5 years	197 (39.8)	109 (38.3)	88 (41.9)		160 (43.8)	37 (28.4)	
>5 years	108 (21.8)	46 (16.1)	62 (29.5)		87 (23.8)	21 (16.2)	
NO antibiotic prescription in PED	86	81	5		50	36	
Continued an already prescribed therapy	104	55	49		90	14	
Amoxicillin							
N (%)	116	90	26		92	24	
Median daily dose (IQR) [mg/kg/day]	75 (63–82)	74 (59–80)	82 (75–95)	<0.001	75 (63–82)	75 (68–81)	n.s.
Median administration/day (IQR)	3 (3–3)	3 (3–3)	3 (3–3)	n.s.	3 (3–3)	3 (3–3)	n.s.
Median length of therapy (IQR)	7 (7–7)	7 (7–7)	7 (7–8)	n.s.	7 (7–7)	7 (5–7)	<0.001
Amoxicillin -clavulanic acid							
N (%)	147	81	66		135	12	
Median daily dose (IQR) [mg/kg/day]	78 (74–92)	80 (74–88)	78 (738–94)	n.s.	78 (74–92)	84 (77–93)	n.s.
Median administration/day (IQR)	3 (3–3)	3 (3–3)	3 (3–3)	n.s.	3 (3–3)	3 (3–3)	n.s.
Median length of therapy (IQR)	7 (7–8)	7 (7–8)	7 (7–8)	0.021	7 (7–8)	6.5 (5.3–8)	0.030
Oral cephalosporin							
N (%)	23	19	4		20	3	
Median daily dose (IQR) [mg/kg/day]	8 (8–8)	8 (8–8)	8 (8–10)	n.s.	8 (8–9)	8 (7–8)	n.s.
Median administration/day (IQR)	2 (2–2)	2 (2–2)	2 (2–2)	n.s.	2 (2–2)	2 (2–2)	n.s.
Median length of therapy (IQR)	7 (7–8)	7 (7–8)	7 (7–7.8)	n.s.	7 (7–8)	7 (6–7)	n.s.
Parenteral cephalosporin							
N (%)	68	6	62		56	12	
Median daily dose (IQR) [mg/kg/day]	57.5 (48–67)	52.5 (44–81)	59 (48–67)	n.s.	55 (45–65)	68 (64–77)	0.004
Median administration/day (IQR)	1 (1–1)	1 (1–1.3)	1 (1–1)	n.s.	1 (1–1)	1 (1–1)	n.s.
Median length of therapy (IQR)	5 (3–6)	3 (1.8–5.5)	5 (3–6)	n.s.	5 (3–6.8)	5 (2.3–5)	n.s.
Clarithromycin							
N (%)	53	41	12		44	9	
Median daily dose (IQR) [mg/kg/day]	16 (15–19)	17 (15–19)	16 (15–18)	n.s.	16 (15–19)	16 (16–19)	n.s.
Median administration/day (IQR)	2 (2–2)	2 (2–2)	2 (2–2)	n.s.	2 (2–2)	2 (2–2)	n.s.
Median length of therapy (IQR)	7 (7–10)	7 (7–10)	7 (7–10)	n.s.	7.5 (7–10)	7 (4–8.5)	0.043
Azithromycin							
N (%)	11	9	2		9	2	
Median daily dose (IQR) [mg/kg/day]	10 (10–10)	10 (10–11)	10 (10–10)	n.s.	10 (10–10)	12 (11–12)	n.s.
Median administration/day (IQR)	1 (1–1)	1 (1–1)	1 (1–1)	n.s.	1 (1–1)	1 (1–1)	0.036
Median length of therapy (IQR)	5 (3–5)	5 (3–5)	5 (5–5)	n.s.	5 (3–5)	3 (3–3.5)	n.s.

**Table 3 life-13-01922-t003:** Univariate logistic regression and multivariate logistic regression with Odds Ratio, 95% Confidence Intervals (CI), and *p*-value in relation to hospital admission. SpO2 = saturation of peripheral oxygen, WBC = white blood cells, N% = WBC percentage of neutrophils, CRP = C-Reactive Protein, CXR = chest X-ray, bpm = beats per minute, n.s. = not significant.

	Univariate			Multivariate		
	OR	CI 95%	*p*	OR	CI 95%	*p*
Female	0.212	0.877–1.806	n.s.			
Age groups (ref < 2 years)						
2–5 years	0.325	0.213–0.496	<0.001	0.369	0.199–0.684	0.002
>5 years	0.347	0.206–0.585	<0.001	0.394	0.164–0.948	0.038
Body temperature > 38.5 °C	1.350	0.912–1.996	n.s.			
Heart rate > 160 bpm	3.194	2.156–4.730	<0.001	2.388	1.365–4.179	0.002
Respiratory rate > 60 breaths per minute	3.583	1.819–7.058	<0.001	1.912	0.906–4.043	n.s.
SpO2 < 92%	7.416	3.213–17.116	<0.001	3.119	1.050–9.266	0.041
N% > 75	1.916	0.918–3.999	n.s.			
CRP > 3 mg/dl	1.549	0.765–3.134	n.s.			
Diagnosis of LRTI	1.042	0.718–1.511	n.s.			
>3 days of fever	0.862	0.601–1.238	n.s.			

**Table 4 life-13-01922-t004:** Univariate logistic regression and multivariate logistic regression with Odds Ratio, 95% Confidence Intervals (CI), and *p*-value in relation to any antibiotic prescription. SpO2 = saturation of peripheral oxygen, WBC = white blood cells, N% = WBC percentage of neutrophils, CRP = C-Reactive Protein, CXR = chest X-ray, bpm = beats per minute, n.s. = not significant.

	Univariate			Multivariate		
	OR	CI 95%	*p*	OR	CI 95%	*p*
Female	1.025	0.648–1.620	n.s.			
Age groups (ref < 2 years)						
2–5 years	2.814	1.642–4.822	<0.001	1.914	0.902–4.061	n.s.
>5 years	4.050	1.863–8.806	<0.001	3.115	0.828–11.721	n.s.
Body temperature > 38.5 °C	1.421	0.823–2.452	n.s.			
Heart rate > 160 bpm	0.514	0.317–0.831	0.007	0.941	0.469–1.888	n.s.
Respiratory rate > 60 breaths per minute	0.039	0.222–0.963	0.039	0.651	0.286–1.485	n.s.
SpO2 < 92%	0.838	0.311–2.260	n.s.			
N% > 75	5.542	0.688–44.672	n.s.			
CRP > 3 mg/dL	7.747	0.977–61.402	n.s.			
Diagnosis of LRTI	0.084	0.033–0.210	<0.001	0.069	0.016–0.297	<0.001
>3 days of fever	4.267	2.460–7.401	<0.001	3.633	1.752–7.533	<0.001

## Data Availability

The data presented in the study are available from the corresponding author upon reasonable request.

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
