# Peer review of "Antibiotic Prescribing for Lower Respiratory Tract Infections and Community-Acquired Pneumonia: An Italian Pediatric Emergency Department’s Real-Life Experience"

_life, 2023, doi:10.3390/life13091922_

Round 1
Reviewer 1 Report
Thankyou for the opportunity to review your manuscript describing paedaitric antibiotic prescribing in the ED for LRTI at a single centre.
Introduction. concise
Materials and Methods.
It would be useful to have a general understanding of your ED - presentations per year, etc as well as local resistence patterns to understand later how to interpret antibiotic prescribing. A large part of your discussion relies on how you implement (or dont implement) guidelines - an understanding of what education and processes around antiobitic prescrining at your facility is useful eg electronic medical record with built in clinical decision rules, paper based pathways, protocols, audit cycles, regular teaching etc.
Noted that you used > 37.5C as an inclusion criteria - could this potential miss a cohort of patients who were also given antibiotics (with even less reason to)? Why was the temperature an inclusion criteria? The data is 6 years old. Why was a more recent cohort not used?
There is no mention about an ethical review process and approval.
Statistical Analysis. fine
Results. How many met the inclusion criteria and how many were excluded due to those criteria?
Table 1 first row is "LRIT" - it should read LTRI
Table 3 & 4 HR > 160 and RR > 60 seem like crude numbers given variability depending on age. Is this relevant and helpful to the interpretation? I'm of the opinion not.
Discussion. This paper is an incidence study of antibiotic prescribing in a specific population. It over reaches when attempting to define parameters for deciding when to give antibiotics in this cohort. As a single centre observational study, these conclusions are hazardous. In addition, there are large multicentred prospective clinical trials such as ARTIC PC to answer these questions. If your study was to go beyond the incidence and appropriateness of antibiotic prescribing, would be to understand why this is the case (ie barriers to implementation - is it education, quality programs, seniority of clinician prescribing etc) - but i think this is much beyond the scope of this paper. Your last sentence in the discussion could go into the conclusion.
Limitations.Individual resistence was not measured and therefore choice of antibiotic may have been correct. Equally, patient allergies were not recorded and therefore it is further unclear how many of the second line agents were in fact appropriate.
english quality high - minor grammatical errors only
you have written die throughout, i think you mean day?
Author Response
Thank you for the opportunity to review your manuscript describing paedaitric antibiotic prescribing in the ED for LRTI at a single centre.
Reply: Thank you for spending time in revising our paper!
Introduction. Concise
Reply: We added more information about local resistance patterns and local antibiotic prescription in children
Materials and Methods.
It would be useful to have a general understanding of your ED - presentations per year, etc as well as local resistence patterns to understand later how to interpret antibiotic prescribing.
Reply: Thanks to your suggestion, we included in this section more information about our PED (methods, page 3) whereas information about local resistance patterns has been added in the introduction.
A large part of your discussion relies on how you implement (or dont implement) guidelines - an understanding of what education and processes around antiobiotic prescrining at your facility is useful eg electronic medical record with built in clinical decision rules, paper based pathways, protocols, audit cycles, regular teaching etc.
Reply: We collected these data as part of a clinical audit in 2018, when there were no particular strategies to evaluate antibiotic prescribing
Noted that you used > 37.5C as an inclusion criteria - could this potential miss a cohort of patients who were also given antibiotics (with even less reason to)? Why was the temperature an inclusion criteria? The data is 6 years old. Why was a more recent cohort not used?
Reply: Fever is considered a prerequisite for bacterial infections, so prescribing antibiotics to non-febrile patients would be grossly wrong. We considered the aforementioned fever threshold to keep the sample more homogeneous as specified in the materials and methods section, page 2. About the period of study, we collected data as part of a clinical audit carried out in the interest of improving the quality of health care.
There is no mention about an ethical review process and approval.
Reply: The data was collected anonymously as part of a clinical audit carried out in the interest of improving the quality of health care, in agreement with the representatives of the reference health facility; therefore, it was not submitted to the opinion of the Ethics Committee. We added this sentence at the end of materials and methods section.
Statistical Analysis. Fine
Reply: thank you.
Results. How many met the inclusion criteria and how many were excluded due to those criteria?
Reply: In Material and Methods one figure about study flow chart was added to show how many patients met the inclusion criteria.
Table 1 first row is "LRIT" - it should read LTRI
Reply: thank you for spotting this inaccuracy, it was corrected.
Table 3 & 4 HR > 160 and RR > 60 seem like crude numbers given variability depending on age. Is this relevant and helpful to the interpretation? I'm of the opinion not.
Reply: We have considered these heart rate and respiratory rate thresholds as they are recognised as alarm parameters in the considered ages by the guidelines of the European Resuscitation Council. The relevance of this categorization is evident in the univariate analyses, as it is relevant that these variables lose significance at the multivariate logistic regression.
Discussion. This paper is an incidence study of antibiotic prescribing in a specific population. It over reaches when attempting to define parameters for deciding when to give antibiotics in this cohort. As a single centre observational study, these conclusions are hazardous. In addition, there are large multicentred prospective clinical trials such as ARTIC PC to answer these questions. If your study was to go beyond the incidence and appropriateness of antibiotic prescribing, would be to understand why this is the case (ie barriers to implementation - is it education, quality programs, seniority of clinician prescribing etc) - but i think this is much beyond the scope of this paper. Your last sentence in the discussion could go into the conclusion.
Limitations.Individual resistence was not measured and therefore choice of antibiotic may have been correct. Equally, patient allergies were not recorded and therefore it is further unclear how many of the second line agents were in fact appropriate.
Reply: The last sentence in the discussion was moved into conclusion. We then added the ARTIC PC study to the discussion and added the limitations highlighted by the reviewer.
you have written die throughout, I think you mean day?
Reply: that’s what we meant. The word was corrected into “day” throughout the paper.
Reviewer 2 Report
This is a tremendously insightful manuscript regarding antibiotic prescription and usage in Italian Pediatric Emergency Units, well-written and with remarkable honesty. Both germ circulation and antibiotic stewardship are important issues in Pediatrics and the authors’ preoccupation towards them is salutary. The conclusions of their work are indeed worrisome and should provide further evidence to implement healthcare programs that ensure better adherence to treatment guidelines. The minor points I wish to make are as follows:
- The second paragraph of subsection 2.1 should be more structured, when it comes to definitions, and I am not entirely sure it belong there and not in the Introduction section
- On Table 1, I would place the last two rows after the first three, as they are all about subject numbers and percentages. Also, I would keep the same number of decimals on each line, for ease of reading (e.g. Mean duration of fever: 3.1 (2.5)/2.9(2.6)/3.4(2.5))
- Table 2 is a bit crowded – should it be organized horizontally? As previous, keep the same number of decimals when referring to the same data set (when discussing the median dose of AmoxiClav, it seems pointless to use an IQR of 73.75-94.25, instead of 74-94, especially since it is not even statistically significant). Also, please change ”die” to ”day” or ”d” on all lines.
- On Tables 3 and 4, is ”apm” a standard abbreviation when referring to Respiratory rate? Also, change PCR to CRP
- Even though ”fast breathing pneumonia” is a thing in itself, I would replace ”fast breathing” on rows 236 and 240 with ”tachypnea” and also ”chest indrawings” on row 236 with ”chest retractions”
- The limitations of the study are correctly identified, especially when it comes to excluded patients – the authors do not mention if their number might have impacted the results.
None
Author Response
This is a tremendously insightful manuscript regarding antibiotic prescription and usage in Italian Pediatric Emergency Units, well-written and with remarkable honesty. Both germ circulation and antibiotic stewardship are important issues in Pediatrics and the authors’ preoccupation towards them is salutary. The conclusions of their work are indeed worrisome and should provide further evidence to implement healthcare programs that ensure better adherence to treatment guidelines.
Reply: thank you for appreciating our work and for spending your time in revising it
The minor points I wish to make are as follows:
- The second paragraph of subsection 2.1 should be more structured, when it comes to definitions, and I am not entirely sure it belong there and not in the Introduction section
Reply: Thanks for this comment, we made a specification about the definitions in Material and Methods.
- On Table 1, I would place the last two rows after the first three, as they are all about subject numbers and percentages. Also, I would keep the same number of decimals on each line, for ease of reading (e.g. Mean duration of fever: 3.1 (2.5)/2.9(2.6)/3.4(2.5))
Reply: thank you, we moved the last three lines where you suggested and we kept only one decimal number.
- Table 2 is a bit crowded – should it be organized horizontally? As previous, keep the same number of decimals when referring to the same data set (when discussing the median dose of AmoxiClav, it seems pointless to use an IQR of 73.75-94.25, instead of 74-94, especially since it is not even statistically significant). Also, please change ”die” to ”day” or ”d” on all lines.
Reply: The table has been arranged on a horizontal page, and the decimals have been corrected as suggested. The term die has been replaced with day.
- On Tables 3 and 4, is ”apm” a standard abbreviation when referring to Respiratory rate? Also, change PCR to CRP
Reply: thank you. The abbreviation apm has been replaced by the more correct breaths per Minute and CRP was corrected.
- Even though ”fast breathing pneumonia” is a thing in itself, I would replace ”fast breathing” on rows 236 and 240 with ”tachypnea” and also ”chest indrawings” on row 236 with ”chest retractions”
Reply: thank you, we changed “indrawings” with “retractions” and “fast breathing” with “tachypnea” where suitable.
- The limitations of the study are correctly identified, especially when it comes to excluded patients – the authors do not mention if their number might have impacted the results.
Reply: Thank you for this suggestion. We added a specific sentence at the end of discussion (page 11)